# Water and Nitrogen Regulation Effects and System Optimization for Potato (*Solanum tuberosum* L.) under Film Drip Irrigation in the Dry Zone of Ningxia China

**Yingpan Yang [1], Juan Yin [1,2,3,*], Zhenghu Ma [1], Xiaodong Wei [1], Fubin Sun [1] and Zhen Yang [1]**

1. College of Civil and Hydraulic Engineering, Ningxia University, Yinchuan 750021, China
2. Engineering Research Center for Efficient Utilization of Water Resources in Modern Agriculture in Arid Regions, Yinchuan 750021, China
3. Ningxia Research Center of Technology on Water-Saving Irrigation and Water Resources Regulation, Yinchuan 750021, China
* Correspondence: yj7115@126.com; Tel.: +86-139-9548-3882

**Abstract:** Potatoes require water and fertilizer management systems to optimize economic outcomes and fertilizer use, especially in arid areas such as Ningxia, China. In this study, three irrigation quotas (1200 (W1), 1650 (W2), and 2100 (W3) $m^3/hm^2$) and three nitrogen application treatments (110 (N1), 190 (N2), and 270 (N3) $kg/hm^2$) were evaluated. Potato growth, final yield, and quality indices were assessed in relation to fertility periods, irrigation utilization efficiency, nitrogen and fertilizer bias productivity, and economic efficiency, using the TOPSIS model. Stem thickness and plant height varied significantly with irrigation and nitrogen treatments; total yield and vitamin C, reducing sugar, and starch contents were the highest under the W3N1, W3N1, W1N2, and W2N3 treatments, increasing by 54.56, 17.00, 69.00, and 45.00%, respectively, compared with those in the control. The regression relationships between water and nitrogen regulation and yield, irrigation water use efficiency, nitrogen fertilizer bias productivity, and economic efficiency agreed with the binary quadratic regression model, and the coefficients of determination ($R^2$) were >0.85. W3N1 was optimal for model yield, nitrogen fertilizer bias productivity, and economic efficiency, and W1N3 was optimal for irrigation water use efficiency. Our findings will help optimize potato management in central Ningxia.

**Keywords:** potato; water and nitrogen regulation; growth; yield; water and nitrogen use efficiency; economic benefits

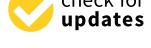



## 1. Introduction

Potato is one of the four staple food crops in China [1]. Therefore, the potato industry plays an important role in ensuring food security in China and protecting the ecological environment [2]. Water and nitrogen supplies are the two major factors limiting crop productivity, including that of potatoes [3], and are key determinants of grain yield [4]. Current trends in potato yield are insufficient to meet the needs of the doubling in food demand expected by 2050 [5]. Furthermore, crop yield losses owing to water scarcity are an increasing threat to agriculture worldwide [6], and yield improvements are dependent on nitrogen accumulation. Thus, nitrogen fertilization research is essential to improve crop yields and environmental viability [7,8]. Furthermore, an integrated consideration of optimizing the synergistic effects of irrigation and nitrogen fertilization is key to achieving sustainable agricultural goals.

Since 1980, nitrogen fertilizer application in China has increased dramatically, and the proportion of nitrogen input has now reached levels that diminish crop yield returns. Nitrogen use efficiency in crop production is crucial to sufficiently address food security, soil degradation, and climate change [9]. Water and fertilizer application are important factors

for crop growth; specifically, large nitrogen inputs during crop growth can help improve grain yield and quality [10–12]. Drip irrigation coupled with water-fertilizer integration can significantly improve crop yield relative to conventional water-fertilizer treatments [13]. Sub-membrane drip irrigation is a common agricultural irrigation method practiced in oasis agroecosystems in northwest China's arid regions, as it can improve agricultural production by suppressing soil evaporation [14]. Previous studies have demonstrated that appropriate irrigation and nitrogen management practices can help obtain high yields in wheat [15]. Potato yield, water conservation, and increased potato water use efficiency can be effectively achieved by simulated irrigation methods compared to automatic irrigation methods in the field [16]. Lower N application reduces potato tuber yield, and appropriate N application and subsurface drip irrigation can result in higher biomass, yield, and water use efficiency [17]. Appropriate limited single irrigation and the control of the soil water content levels can increase the total yield, water use efficiency, and the leaf area index [18]. Miguel et al. [19] determined that water and nitrogen have strong interactions in agricultural systems by studying methods to simultaneously improve water and nitrogen use efficiency. Management practices that aim to reduce nitrogen losses and maintain farm productivity should thus optimize both nitrogen and water use. Sadras et al. [20] analyzed the interactions between water and nitrogen from physiological, agronomic, economic, breeding, and modeling perspectives by comparing phenotypic changes in wheat in Australia, the UK, Argentina, and Italy. Their results showed that increased N uptake was correlated with yield increases and helped preserve grain protein content. High levels of water irrigation combined with N application resulted in increased cotton yields and water productivity [21]. Optimal water-N patterns can also be used to achieve high rice yields and reduce N losses [22]. Low/high-frequency N application combined with water-saving irrigation can significantly reduce water and N availability to achieve food security and efficient resource use in rice production while maintaining stable yields [23]. Therefore, understanding the water and nitrogen coupling relationship is key for producing high-quality potatoes with greater yields while maintaining efficient use of water and fertilizer resources. To achieve this, further research is required to elucidate the response mechanisms of potatoes to water and nitrogen regulation.

Previous studies investigating optimal water and nitrogen models only utilized certain indicators, as balancing the multiple objectives involved with water and fertilizer efficiency, yield, and economic benefits is challenging. Therefore, this aspect requires further investigation. Moreover, studies investigating the multi-objective optimization of water and fertilizer regulation in relation to potato production, economic benefits, and environmental benefits in the semi-arid areas of Ningxia are limited.

Therefore, in this study, we investigated the effects of water and nitrogen regulation on potato growth, yield, quality, water use efficiency, nitrogen fertilizer bias productivity, and economic efficiency based on local conditions and described and quantified the responses to the different factors examined. A mathematical model was constructed based on water and nitrogen regulation, and each index was based on a binary quadratic regression model. Following this, the model was optimized using the TOPSIS model to analyze each index and determine the optimal amounts of irrigation water and nitrogen application. The results of this study will provide a scientific theoretical basis and practical guidance to achieve high yields while reducing water and fertilizer application in potato cultivation under film drip irrigation in the arid areas of Ningxia, China.

## 2. Materials and Methods

### 2.1. Experimental Site

Field experiments were conducted in 2021 and 2022 at a site located in Yuwang Town, Concentric County, Wuzhong City, Central Ningxia, China, a typical region in this arid zone ($36°48'2''$ N, $106°21'53''$ E, altitude 1489 m), as shown in Figure 1a. The area has an arid continental climate, with large temperature differences between day and night, year-round drought, low levels of rain, long light hours, strong surface evaporation, and

groundwater below 10 m, with an average rainfall of approximately 270 mm and annual evaporation of approximately 2325 mm from 2012 to 2022. Rainfall and temperature during the potato growing season are shown in Figure 2. The soil at the test site was defined as sandy loam, and the basic physical and chemical properties of the soil in the 0–20 cm tillage layer before sowing were as follows [24]: bulk weight, 1.41 g/cm$^3$; field water holding capacity, 21.8%; pH, 8.57; total salt, 0.6 g/kg; organic matter, 6.65 g/kg; alkaline nitrogen, 38 mg/kg; fast-acting phosphorus, 3.94 mg/kg; fast-acting potassium, 130 mg/kg; total nitrogen, 0.027%. The total nitrogen, phosphorus, and potassium contents were 0.027, 0.064, and 1.74%, respectively.

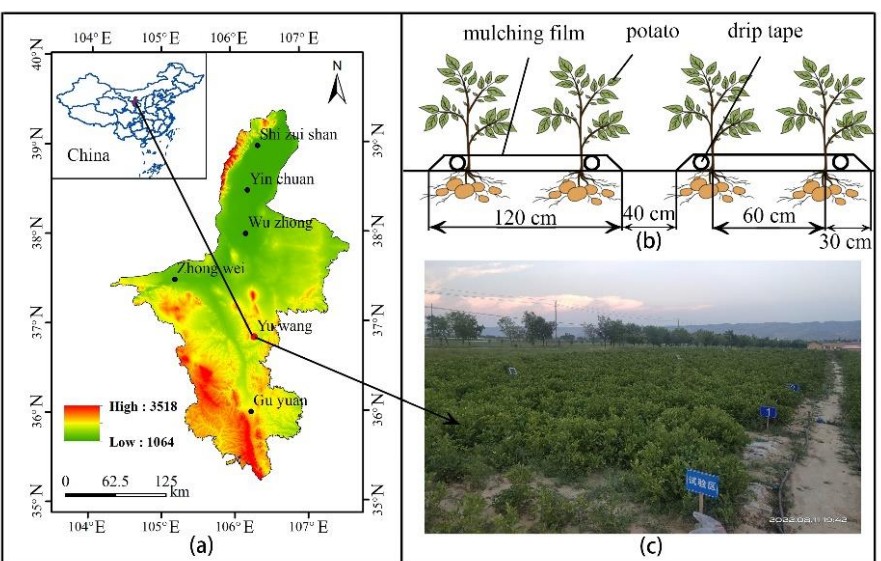

**Figure 1.** (**a**) Location of the experimental site, (**b**,**c**) Potato planting pattern and drip irrigation belt arrangement diagram.

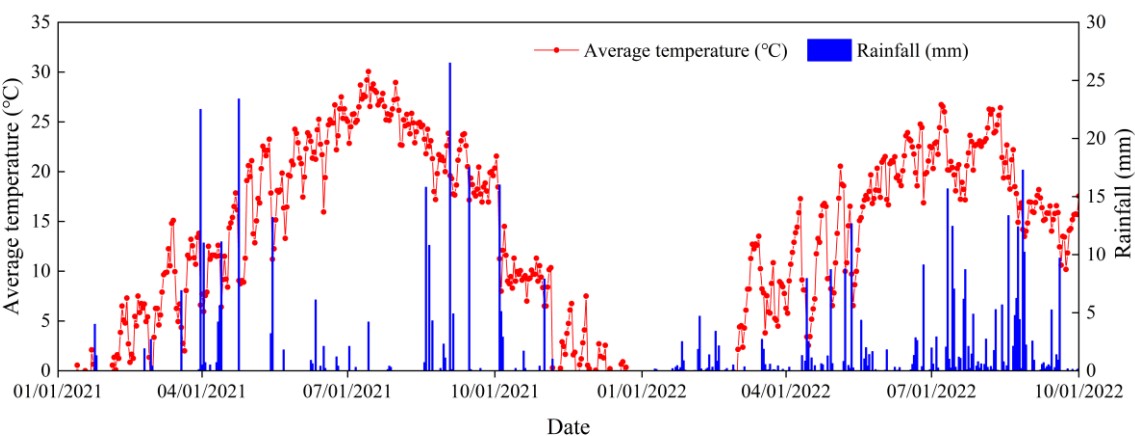

**Figure 2.** Temperature and rainfall during the potato fertility period in the experimental area.

## 2.2. Experimental Design

The irrigation quota for under-membrane drip irrigation was set by referencing the "Ningxia Potato Drip Irrigation Planting Technology Regulations" issued by the Department of Water Resources of Ningxia Hui Autonomous Region in 2017 [25]. Three drip irrigation levels and three nitrogen application levels were designed, and the actual local production without irrigation and N application was used as the control group CK. The irrigation treatments were as follows: W3, 2100 m$^3$·hm$^{-2}$; W2, 1680 m$^3$·hm$^{-2}$ (water saving: 20%); and W1, 1260 m$^3$·hm$^{-2}$ (water saving: 40%). The N application treatments were as follows: N3, 270 kg·hm$^{-2}$; N2, 190 kg·hm$^{-2}$ (30% N reduction); and N1, 110 kg·hm$^{-2}$ (60%

N reduction). The specific irrigation and N application amounts are shown in Figure 3. Calcium superphosphate (12% phosphorus, 82.5 kg/kg/hm$^2$) and potassium sulfate (50% potassium, 150 kg/kg/hm$^2$) were selected as the basal applications, and nitrogen fertilizer was applied multiple times using water–fertilizer integration (nitrogen fertilizer is 46% urea). The chasing period was carried out six times according to the seedling stage, tuber formation stage, tuber growth stage, and starch accumulation stage (1:2:2:1). Irrigation was carried out 10 times according to the shoot growth period, seedling stage, tuber formation stage, tuber growth stage, and starch accumulation stage (1:2:3:3:1), as shown in Table 1.

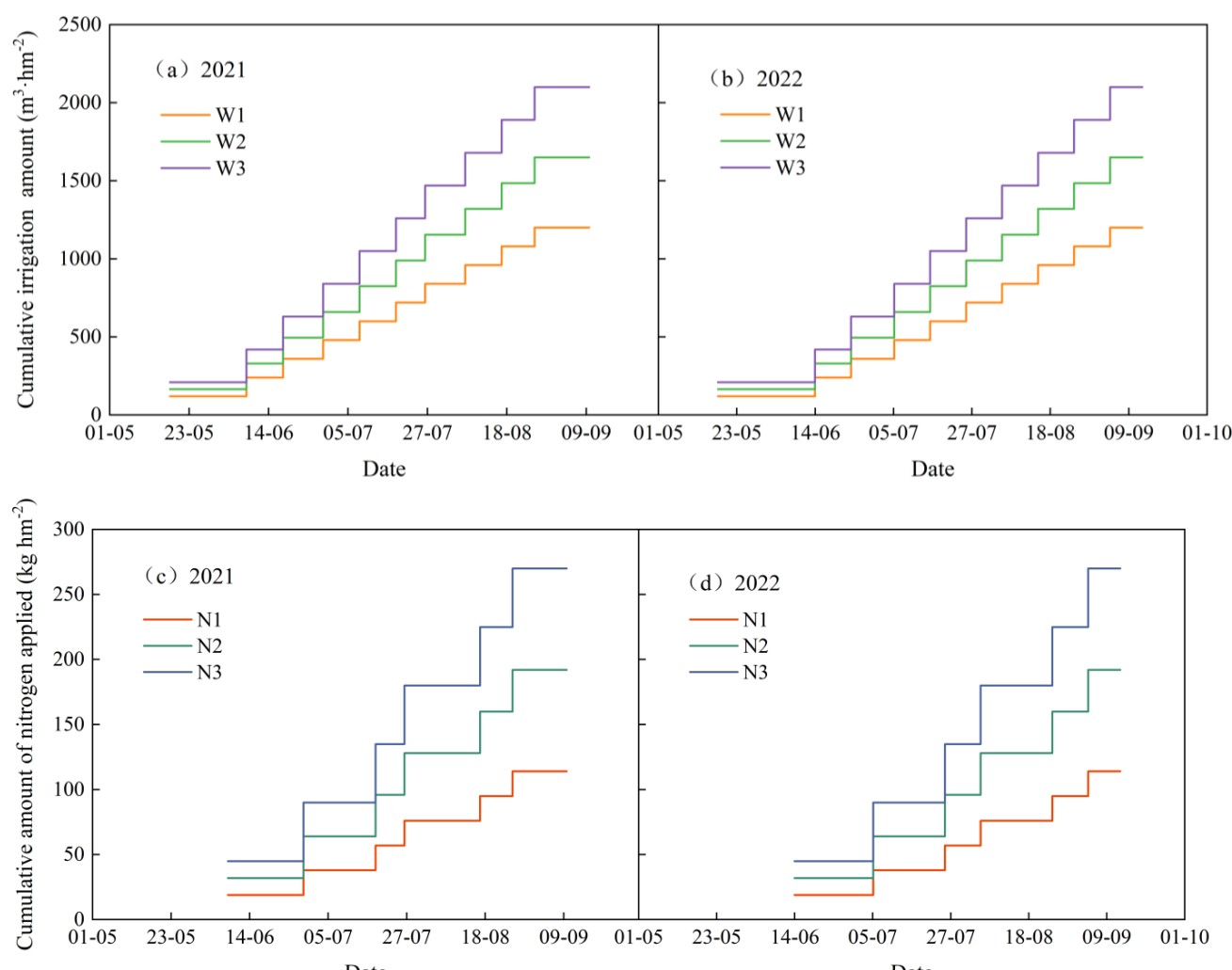

**Figure 3.** N accumulation with each application and irrigation treatment during potato fertility.

**Table 1.** Allocation ratio of the irrigation and N applications during the potato fertility periods.

| Potato Growth | Bud Growth Period (5.10–6.5) | Seedling Stage (6.6–6.25) | Tuber Formation Period (6.26–7.25) | Tuber Growth Period (7.26–8.20) | Starch Accumulation Period (8.21–10.5) | Total |
|---|---|---|---|---|---|---|
| Percentage of irrigation water | 8.85% (1) | 17.7% (2) | 31.86% (3) | 32.74% (3) | 8.85% (1) | 100% |
| Percentage of nitrogen application | 0 | 16.7% (1) | 33.3% (2) | 33.3% (2) | 16.7% (1) | 100% |

Note: The number of irrigation and N applications per growth stage is indicated in parentheses.

Each treatment was set up with three replications for a total of 30 plots. The potato variety 'qingshu9' was tested in the experiment; 'qingshu9' is a potato variety bred by the Institute of Biotechnology, Qinghai Academy of Agricultural and Forestry Sciences by crossing 3875213 × APHRODITE; it is drought-resistant, cold-resistant, and resistant to late blight and ring rot. Drip irrigation planting mode was applied, as shown in Figure 1b,c. Seed potatoes were sown on 30 April 2021 and 1 May 2022 and planted with a distance of 50 cm between plants and 60 cm between rows, with 10 plants in each row and 40 potatoes in each plot. A planting density of 33,345 plants·ha$^{-1}$ was maintained in each plot. Each plot, with an area of 17.6 m$^2$ (5.5 m × 3.2 m, length × width), was surrounded by protection rows, with the width of the protection rows between the plots being 1 m and that of the outer protection rows being 2 m. Each plot was equipped with an independent branch control unit, including a water meter, gate valve, and pressure gauge. Fertilizer application, pesticide application, and other agronomic measures were performed according to local practices. Detoxified seed potatoes were selected before planting; first, when selecting seed potatoes, old, deformed, damaged, diseased, and rotten potatoes were eliminated, and qualified seed potatoes were germinated and stored in warm indoor conditions prior to sowing. The germination length was 0.5–1 cm. Thereafter, seed potatoes were cut 2–3 d prior to sowing, ensuring that 1–2 bud eyes remained on each cut block. The cutter used for cutting was disinfected with 75% alcohol to prevent the spread of infection during the cutting process. Potatoes were harvested on 2 October 2021 and 3 October 2022.

*2.3. Measurement Items and Methods*

2.3.1. Potato Growth

Three potato plants were selected from different treatments in each plot to represent the overall growth in the plot, and the stem thickness and plant height were measured and averaged. Stem thickness was measured using Vernier calipers with an accuracy of 0.1 mm; plant height was measured using a steel ruler with an accuracy of 1 mm.

2.3.2. Determining Yield and Yield Components

At maturity, potatoes with good growth in a 2 × 2 m area were harvested and randomly boxed from each test plot and weighed on an electronic balance (0.01 g) to evaluate yield composition (classified according to the national standard "GB/T31784-2015 potato commercial potato grading and inspection protocol": small potatoes ≤ 75 g, 75 g < medium potatoes ≤ 150 g, large potatoes > 150 g). The average yield of single plants was calculated, and the total yield of potatoes per hectare was determined for the different treatments according to the planting density.

2.3.3. Potato Quality

To determine potato quality [26], vitamin C (VC), reducing sugar, and starch contents were determined using the 2,6-dichloroindophenol titration, potassium permanganate titration, and acid hydrolysis methods, respectively.

2.3.4. Irrigation Water Use Efficiency

Irrigation water use efficiency (*iWUE*) was defined as follows [27]:

$$iWUE = Y/I \tag{1}$$

where *iWUE* is the irrigation water use efficiency, *Y* is the potato yield (kg/hm$^2$), and *I* is the irrigation water supplied according to the potato fertility stage (m$^3$/hm$^2$).

2.3.5. Nitrogen Fertilizer Biased Productivity

Nitrogen fertilizer biased productivity (*PFP*) was defined as follows [28]:

$$PFP = Y/FT \tag{2}$$

where *PFP* is the fertilizer bias productivity, *Y* is the potato yield (kg/hm$^2$), and *FT* is the total nitrogen fertilizer application (kg/hm$^2$).

### 2.3.6. Economic Benefits

The economic benefits (*EB*) were calculated as described previously [29,30]:

$$EB = YR - IW - NW - OW - PW \tag{3}$$

where *EB* is economic benefit (CHY/hm$^2$), *YR* is the potato yield income (CHY/hm$^2$), *IW* is the irrigation water input (CHY/hm$^2$), *NW* is the nitrogen application input (CHY/hm$^2$), *OW* is other inputs, and *PW* is the manual labor cost (CHY/hm$^2$).

### 2.4. Multi-Objective Decision Making and TOPSIS Evaluation

The technique for order preference by similarity to ideal solution (TOPSIS) was used to identify a solution from the feasible solution set by defining the positive ideal solution and the negative ideal solution for the decision problem so that it was closest to the positive ideal solution and farthest from the negative ideal solution [31].

(1) Nine treatments were set as evaluation objects, with nine evaluation indicators including fruit yield, *iWUE*, *PFP*, and *EB*. The evaluation indicators were normalized to establish a normalized matrix:

$$z_{ij} = \frac{x_{ij}}{\sqrt{\sum_{i=1}^{n} x_{ij}^2}} \tag{4}$$

where $z_{ij}$ is the *j* index normalized value in *i* treatment; $x_{ij}$ is the *j* index value in the *i* treatment. *i* = 1, 2,···, n; *j* = 1, 2,···, m;

(2) The ideal solution ($Z_{ij}^+$) and the negative solution ($Z_{ij}^-$) were determined to form the ideal solution vector $Z^+$ and the negative solution vector $Z^-$, respectively:

$$Z_{ij}^+ = \left( z_{i1}^+, z_{i1}^+, z_{i3}^+ \ldots \ldots z_{ij}^+ \right) \tag{5}$$

$$Z_{ij}^- = \left( z_{i1}^-, z_{i1}^-, z_{i3}^- \ldots \ldots z_{ij}^- \right) \tag{6}$$

where $Z_{ij}^+$ and $Z_{ij}^-$ represent the maximum and minimum values of the evaluation object in the *j*-th index, respectively;

(3) The Euclidean distances ($D_i^+$ and $D_i^-$) were determined:

$$D_i^+ = \sqrt{\sum_{j=1}^{m} \left[ w_j \times \left( z_{ij} - Z_{ij}^+ \right) \right]^2} \tag{7}$$

$$D_i^- = \sqrt{\sum_{j=1}^{m} \left[ w_j \times \left( z_{ij} - Z_{ij}^- \right) \right]^2} \tag{8}$$

where $w_j$ is the weight of indicator *j*;

(4) The relative proximity coefficient $R_i$ of each treatment was calculated; that is, the proximity between the evaluation object and the optimal scheme was calculated as follows:

$$R_i = \frac{D_i^-}{D_i^+ + D_i^-}. \tag{9}$$

### 2.5. Data Analysis

Microsoft Excel 2019 was used for data collation, Origin 2018 for plotting, and SPSS 26.0 for statistical analysis, and Duncan's multiple comparisons were used to test for significance ($p < 0.05$).

## 3. Results

### 3.1. Effect of Water and Nitrogen Regulation

#### 3.1.1. Potato Growth

Figure 4 shows the effects of water and nitrogen regulation on potato growth. We observed that the differences between the treatments over the two years were consistent. The stem thickness of potato plants was significantly ($p < 0.05$) affected by the different water and nitrogen treatments and increased as the fertility period progressed. Under the same irrigation conditions, the stem thickness at the seedling stage showed a trend of increasing and decreasing with increasing N application. Overall, the W3N3 treatment yielded significantly better results with respect to stem thickness compared with the other treatments, showing an increase of 66.04–76.21% compared with the CK. Tuber formation under the W1 irrigation quota followed the same trend as that at the seedling stage, and tuber formation under the W2 and W3 irrigation quotas increased with increasing N application. Moreover, stem thickness was the highest under the W3N3 treatment. The change in the tuber growth stage increased with N application, and no significant change was observed under the W3 irrigation quota; the trends observed at the starch accumulation and maturity stages were consistent, and the stem thickness was the highest under the W3N3 treatment. At the same N application rate, the stem thickness decreased and subsequently increased with increasing irrigation quotas (except at maturity). At maturity, the stem thickness under the N1 and N2 treatments increased with increasing irrigation quotas, whereas that under the N3 treatment showed a trend of increasing and then decreasing owing to the aging potato stalks.

Plant height was significantly affected by the different water and nitrogen treatments ($p < 0.05$). Under the same irrigation conditions, plant height in the W1 treatment at the seedling stage showed an increasing trend, followed by a decreasing trend with increasing N application. Plant height under the W3N2 treatment was the highest at the seedling, tuber growth, and starch accumulation stages, with an average increase of 54.22, 55.70, 62.95, and 49.34% compared to the CK, and plant height under the W3N3 treatment was the highest at the tuber formation and maturity stages. Under the same conditions of nitrogen application, the plant height under each treatment showed a decreasing and a subsequently increasing trend with the increases in irrigation quota at the seedling and tuber growth stages. Plant height under the N1 treatment gradually increased with the irrigation quota at the tuber formation and starch accumulation stages, and that under the N2 and N3 treatments showed a decreasing trend followed by a subsequent increase. Plant height under the N1 treatment gradually increased, that under the N2 treatment initially increased and then decreased, and plant height under the N3 treatment showed a decreasing and subsequently increasing trend at the maturity stage. These results indicate that increased water N treatments could significantly improve potato growth.

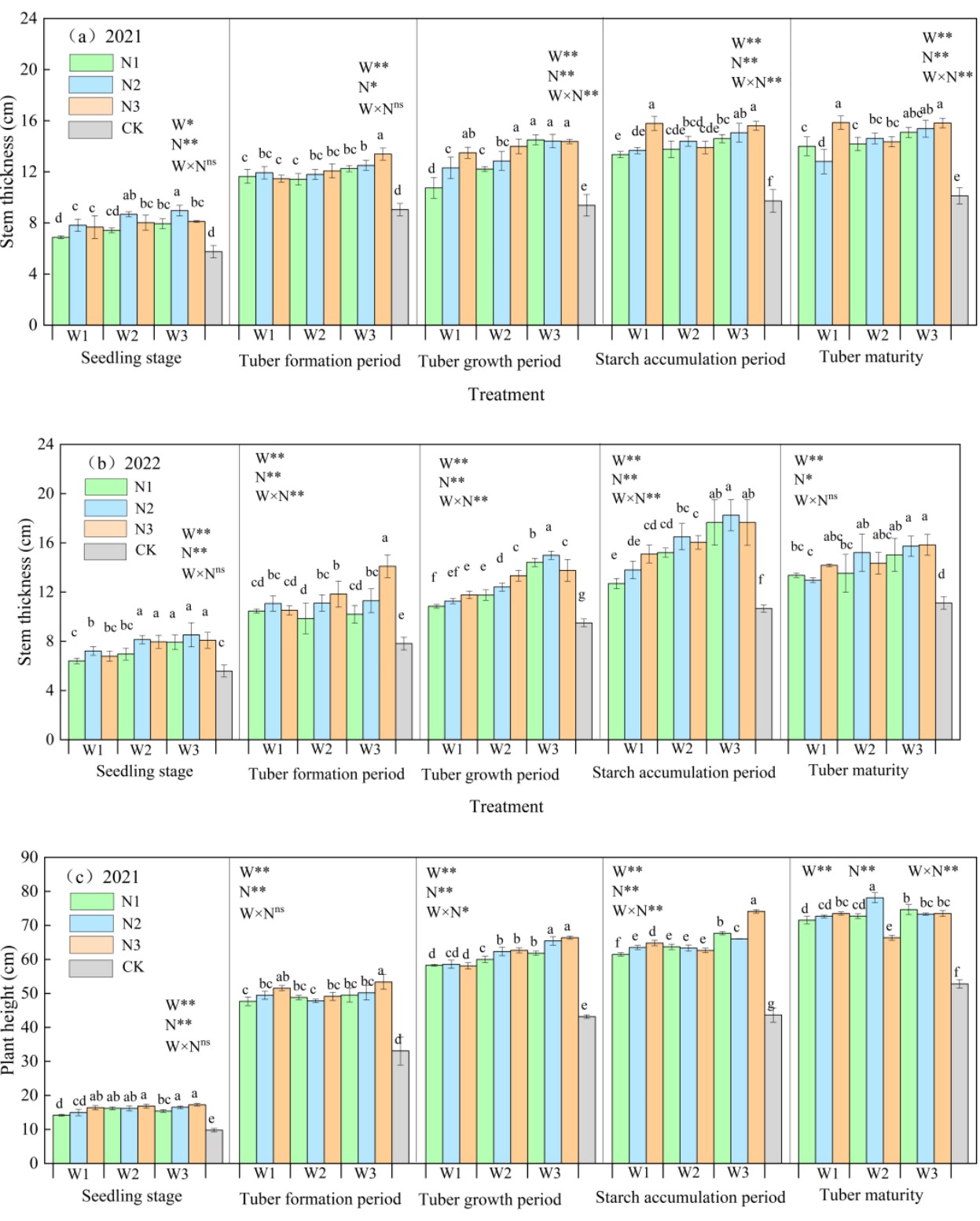

**Figure 4.** *Cont.*

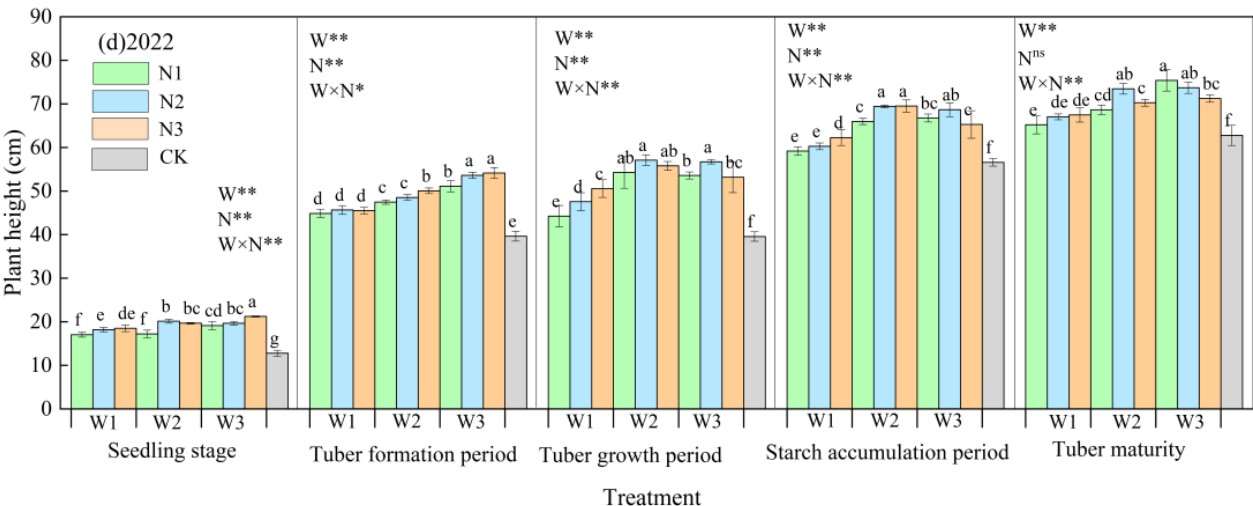

**Figure 4.** Effects of water and nitrogen regulation on potato growth. Note: (**a**) is potato plant stem thickness in 2021, (**b**) is potato plant stem thickness in 2022, (**c**) is potato plant height in 2021 and (**d**) is potato plant height in 2022. * and ** indicate $p < 0.05$ and $p < 0.01$, respectively; ns indicates that there was no significant effect ($p > 0.05$). N is the nitrogen application rate, and W is the irrigation quota.

### 3.1.2. Potato Yield

The effects of the water and N regulation on potato yield are shown in Figure 5. The level of irrigation and N application had a significant effect on the yield components ($p < 0.05$), and the differences between treatments were consistent between the two years. The total potato yield was 39,931.68–54,085.48 kg/hm$^2$, showing an increase of 2139.35–16,293.14 kg/hm$^2$ or 43.11–66.00% compared with that in the CK. Under the same irrigation quota, potato yield under the W1 and W2 treatments showed an increasing trend with increasing N applications, whereas that under the W3 treatment showed a decreasing trend followed by an increasing trend. Under the same N application treatments, potato yield under the N1 and N3 treatments increased with increasing irrigation quotas, whereas that under the N2 treatment showed an increasing and a subsequent decreasing trend. The yield under the W3N1 treatment was the highest, which increased by 52.36% when compared with that in the CK; thus, the highest yield could be obtained with increased irrigation and low nitrogen application, with excessive nitrogen application causing a significant decrease in yield. The yield of large potatoes was the highest under the W3N1 and W3N3 treatments, increasing by 86.22% and 86.67%, respectively, compared with that in the CK. The yield of medium potatoes was the highest under the W1N2 and W3N1 treatments, showing an increase of 49.14% and 55.32%, respectively, compared with that in the CK, whereas the yield of small potatoes was the largest under the W2N3 treatment, increasing by 6.27% compared with that in the CK. Overall, the analysis showed that potato yield was more sensitive to irrigation than N application and that the maximum irrigation quota and N applications were favorable for large potato yield but not for total yield. Therefore, only appropriate irrigation and N application rates are beneficial for potato yield improvement.

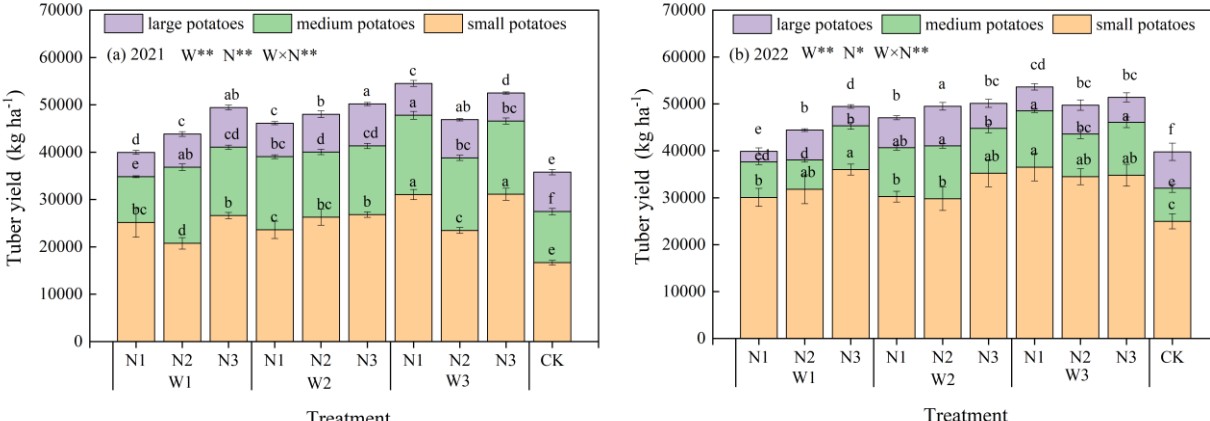

**Figure 5.** Effects of different water and nitrogen treatments on tuber yield. Note: (**a**) is potato tuber yield in 2021 and (**b**) is potato tuber yield in 2022. * and ** indicate $p < 0.05$ and $p < 0.01$, respectively; ns indicates that there was no significant effect ($p > 0.05$). N is the nitrogen application rate, and W is the irrigation quota.

### 3.1.3. Potato Quality

The effects of the different water and nitrogen regulations on potato quality are shown in Figure 6. Potato VC content under the same irrigation quotas under the W1 and W2 treatments initially decreased and then increased with N application, whereas that under the W3 treatment showed a decreasing trend with the same N application. The VC content under the N1 treatment showed a trend of decreasing and then increasing with the irrigation quota, and that under the N2 treatment showed a gradual increase. The VC content under the N3 treatment showed a trend of increasing and then decreasing. The VC content under the W3N1 treatment was the highest, with an average increase of 17% compared with that in the CK. The reducing sugar content significantly differed among treatments, with W1N2 and W3N2 treatments exhibiting significantly higher contents than those observed under the other treatments, being 69% and 39% higher compared with those in the CK, respectively. The potato starch content under the W3N3 treatment was the highest, with an average increase of 45% compared with that in the CK. In summary, all quality indicators were affected differently by the absence of water and nitrogen, indicating that water and nitrogen can significantly improve potato quality.

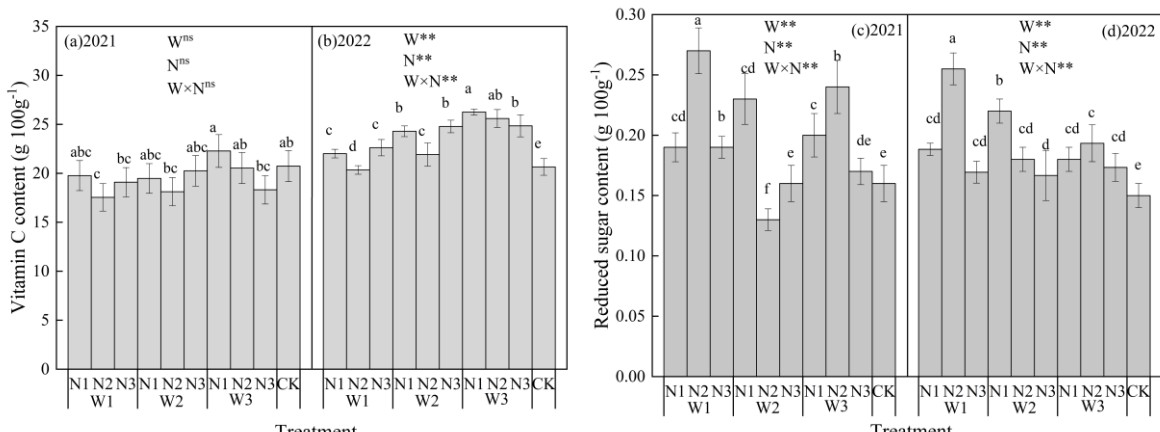

**Figure 6.** *Cont*.

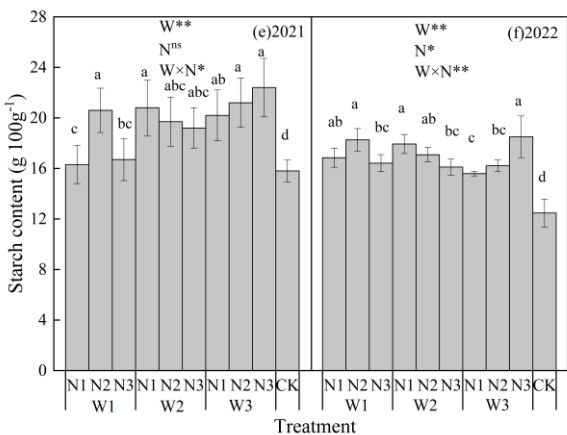

**Figure 6.** Effect of different water and nitrogen levels on quality regulation. Note: (**a**) for potato vitamin C content in 2021, (**b**) for potato vitamin C content in 2022, (**c**) for potato reducing sugar content in 2021, (**d**) for potato reducing sugar content in 2022, (**e**) for potato starch content in 2021, and (**f**) for potato starch content in 2022. * and ** indicate $p < 0.05$ and $p < 0.01$, respectively; ns indicates that there was no significant effect ($p > 0.05$). N is the nitrogen application rate, and W is the irrigation quota.

### 3.2. Effect of Water and Nitrogen Regulation on Irrigation Water Use Efficiency, Nitrogen Fertilizer Bias Productivity, and Economic Efficiency

The effects of water and nitrogen regulation on irrigation water-use efficiency and nitrogen *PFP* are shown in Table 2. Irrigation water-use efficiency decreased with decreasing irrigation quotas at different N applications and was slightly different at the same irrigation quotas; that is, the W1 and W2 treatments exhibited increased *iWUE* with increasing N applications, and the *iWUE* under W3 treatment showed a decreasing and subsequently increasing trend. The highest *iWUE* was found in the W1N3 treatment, which increased by 123% compared with that in the CK. This indicates that the irrigation quota has a greater effect on the irrigation water use efficiency than N application. *PFP* is an important indicator that reflects the combined effects of the local soil base nutrient level and fertilizer dosage. As shown in Table 3, potato N *PFP* was higher under the N1 treatment than that under other treatments at the same irrigation rate, and the highest N *PFP* was 487.64–495.73 kg/kg under the W3N1 treatment, which indicates that the maximum N fertilizer productivity bias and improved N fertilizer utilization can be obtained at a low N application rate.

**Table 2.** Effects of water and nitrogen regulation on irrigation water use efficiency and nitrogen fertilizer bias productivity.

| Year | Treatments | Yield kg·hm$^{-2}$ | Irrigation Volume m$^3$·hm$^{-2}$ | *iWUE* | Total N Fertilizer Application kg | *PFP* kg·kg$^{-1}$ |
|------|-----------|-------------------|-----------------------------------|--------|-----------------------------------|--------------------|
| 2021 | W1N1 | 39,968.72g | 1200 | 33.31g | 110 | 363.35g |
| | W1N2 | 43,842.50f | 1200 | 36.54f | 190 | 230.75f |
| | W1N3 | 49,424.70cd | 1200 | 41.19cd | 270 | 183.05cd |
| | W2N1 | 46,127.25ef | 1560 | 29.57ef | 110 | 419.34ef |
| | W2N2 | 48,006.92cde | 1560 | 30.77cde | 190 | 252.67cde |
| | W2N3 | 50,189.03bc | 1560 | 32.17bc | 270 | 185.89bc |
| | W3N1 | 54,530.74a | 2100 | 25.97a | 110 | 495.73a |
| | W3N2 | 46,868.25de | 2100 | 22.32de | 190 | 246.68de |
| | W3N3 | 52,499.85ab | 2100 | 25.00ab | 270 | 194.44ab |
| | CK | 35,790.30h | 2100 | 17.04h | 150 | 238.60h |

**Table 2.** *Cont.*

| Year | Treatments | Yield kg·hm$^{-2}$ | Irrigation Volume m$^3$·hm$^{-2}$ | *iWUE* | Total N Fertilizer Application kg | *PFP* kg·kg$^{-1}$ |
|---|---|---|---|---|---|---|
| | W1N1 | 39,894.65d | 1200 | 33.25d | 110 | 362.68d |
| | W1N2 | 44,439.21c | 1200 | 37.03c | 190 | 233.89c |
| | W1N3 | 49,459.19ab | 1200 | 41.22ab | 270 | 183.18ab |
| | W2N1 | 46,955.98bc | 1560 | 30.10bc | 110 | 426.87bc |
| 2022 | W2N2 | 49,504.43ab | 1560 | 31.73ab | 190 | 260.55ab |
| | W2N3 | 50,111.87ab | 1560 | 32.12ab | 270 | 185.6ab |
| | W3N1 | 53,640.21a | 2100 | 25.54a | 110 | 487.64a |
| | W3N2 | 49,749.52ab | 2100 | 23.69ab | 190 | 261.84ab |
| | W3N3 | 51,423.99ab | 2100 | 24.49ab | 270 | 190.46ab |
| | CK | 39,794.37f | 2100 | 18.95f | 150 | 265.30f |

Note: Different letters after the values in the same column indicate significant differences between treatments ($p < 0.05$). *iWUE*, irrigation water use efficiency; *PFP*, fertilizer biased productivity.

**Table 3.** Effect of water and nitrogen regulation on economic efficiency.

| Year | Treatment | Water Costs | Fertilizer Costs | Other Costs | Labor Costs | Total Input Costs | Yield Gains | Economic Benefits (CNY/hm$^2$) |
|---|---|---|---|---|---|---|---|---|
| 2021 | W1N1 | 4800.00 | 1955.00 | 847.00 | 2250.00 | 9852.00 | 67,946.82 | 58,094.82 |
| | W1N2 | 4800.00 | 2275.00 | 847.00 | 2250.00 | 10,172.00 | 74,532.25 | 64,360.25 |
| | W1N3 | 4800.00 | 2595.00 | 847.00 | 2250.00 | 10,492.00 | 84,021.99 | 73,529.99 |
| | W2N1 | 6240.00 | 1955.00 | 847.00 | 2250.00 | 11,292.00 | 78,416.33 | 67,124.33 |
| | W2N2 | 6240.00 | 2275.00 | 847.00 | 2250.00 | 11,612.00 | 81,611.77 | 69,999.77 |
| | W2N3 | 6240.00 | 2595.00 | 847.00 | 2250.00 | 11,932.00 | 85,321.35 | 73,389.35 |
| | W3N1 | 8400.00 | 1955.00 | 847.00 | 2250.00 | 13,452.00 | 92,702.26 | 79,250.26 |
| | W3N2 | 8400.00 | 2275.00 | 847.00 | 2250.00 | 13,772.00 | 79,676.03 | 65,904.03 |
| | W3N3 | 8400.00 | 2595.00 | 847.00 | 2250.00 | 14,092.00 | 89,249.75 | 75,157.75 |
| | CK | / | 2115.00 | 847.00 | 2250.00 | 5212.00 | 60,843.51 | 55,631.51 |
| 2022 | W1N1 | 4800.00 | 1955.00 | 847.00 | 2250.00 | 9852.00 | 67,820.90 | 57,968.90 |
| | W1N2 | 4800.00 | 2275.00 | 847.00 | 2250.00 | 10,172.00 | 75,546.67 | 65,374.67 |
| | W1N3 | 4800.00 | 2595.00 | 847.00 | 2250.00 | 10,492.00 | 84,080.63 | 73,588.63 |
| | W2N1 | 6240.00 | 1955.00 | 847.00 | 2250.00 | 11,292.00 | 79,825.17 | 68,533.17 |
| | W2N2 | 6240.00 | 2275.00 | 847.00 | 2250.00 | 11,612.00 | 84,157.53 | 72,545.53 |
| | W2N3 | 6240.00 | 2595.00 | 847.00 | 2250.00 | 11,932.00 | 85,190.17 | 73,258.17 |
| | W3N1 | 8400.00 | 1955.00 | 847.00 | 2250.00 | 13,452.00 | 91,188.36 | 77,736.36 |
| | W3N2 | 8400.00 | 2275.00 | 847.00 | 2250.00 | 13,772.00 | 84,574.18 | 70,802.18 |
| | W3N3 | 8400.00 | 2595.00 | 847.00 | 2250.00 | 14,092.00 | 87,420.79 | 73,328.79 |
| | CK | / | 2115.00 | 847.00 | 2250.00 | 5212.00 | 67,650.42 | 62,438.42 |

Note: Irrigation water, 4 CNY m$^{-1}$; urea, 4 CNY kg$^{-1}$; calcium superphosphate, 2 CNY kg$^{-1}$; potassium sulfate, 9 CNY kg$^{-1}$; potato labor cost, 150 CNY mu$^{-1}$; potato Ningxia market average price, 1.7 CNY kg$^{-1}$.

The economic benefits of each treatment were analyzed by calculating the inputs and benefits of each treatment according to the local economic conditions (Table 3). The highest economic benefit was observed under the W3N1 treatment, with an increase of 33.48% compared with that in the CK. Under the same irrigation quota, the economic benefits of N1 and N2 showed an increasing trend, whereas N3 showed an increasing and then decreasing trend, indicating that appropriate input applications of N could increase the economic benefits. In the N1 treatment, the economic efficiency increased with the irrigation quota, which indicated that the irrigation amount could increase the economic efficiency with low N application; in the N2 treatment, the economic efficiency increased with the irrigation quota and then decreased, which indicated that the economic efficiency decreased with increased irrigation. In contrast, the economic efficiency did not change significantly under the high water and high fertilizer treatments. In summary, examining the three different water and nitrogen treatments in the experimental design, a moderate increase in water and fertilizer inputs was conducive to improving economic benefits; in actual production, farmers can reduce the water and nitrogen inputs, which will lead to more substantial economic benefits.

### 3.3. Interaction Effects of Water and Nitrogen Regulation on Yield, Irrigation Water Use Efficiency, Nitrogen and Fertilizer Bias Productivity, and Economic Efficiency

Based on the data obtained, four models were developed to predict yield (Y), *iWUE*, *PFP*, and *EB* with irrigation quotas and N application as dependent variables and irrigation water (W) and pure nitrogen application (N) as independent variables. A regression analysis was conducted to establish a binary quadratic regression equation, as shown in Table 4, which shows the relationship of irrigation and N application with potato yield, *iWUE*, N *PFP*, and economic efficiency. All $R^2$ values were > 0.85, and the predicted values were significantly correlated with the measured values; therefore, the four models could better predict the changes in water and N regulation for each index.

**Table 4.** Regression relationship between water and nitrogen regulation and yield, irrigation water use efficiency, nitrogen fertilizer bias productivity, and economic efficiency.

| Year | Dependent Variable | Regression Equation | $R^2$ |
|---|---|---|---|
| 2021 | Y | $Y_1 = 6902.82475 + 38.62586166W - 0.004839012346W^2 + 0.3996197557N^2 - 0.07896940836WN$ | 0.890 |
| | *iWUE* | $iWUE_1 = 52.9508439 - 0.021943699310W + 0.03813119892N + 0.000005859315017W^2 + 0.00020698097448N^2 - 0.00005985044414WN$ | 0.984 |
| | *PFP* | $PFP_1 = 457.353369 + 0.31334747347W - 3.912016346N - 0.000028904191564W^2 + 0.009938078578N^2 - 0.0008358779081WN$ | 0.988 |
| | EB | $EB_1 = 6690.11241 + 61.76638016W - 0.008226316872W^2 + 0.6712452012N^2 - 0.13478706255WN$ | 0.851 |
| 2022 | Y | $Y_2 = -9095.38809 + 47.76779682W + 111.33762089N - 0.007494790809W^2 + 0.10675963542N^2 - 0.08026080958WN$ | 0.956 |
| | *iWUE* | $iWUE_2 = 43.8408005 - 0.016293068416W + 0.09862280702N + 0.000004180384088W^2 + 0.00004739583333N^2 - 0.00006049890351WN$ | 0.989 |
| | *PFP* | $PFP_2 = 367.841533 + 0.3772425026W - 3.398549616N - 0.00005005829904W^2 + 0.008434114583N^2 - 0.0008037920322WN$ | 0.994 |
| | EB | $EB_2 = -20074.09519 + 77.20515958W + 185.27410855N - 0.012741114540W^2 + 0.18149114583N^2 - 0.13644341192WN$ | 0.937 |

Note: *EB*, economic benefits; *Y*, yield.

The corresponding values of the water and nitrogen variables with the maximization of each indicator were calculated using the binary quadratic regression equation established for each indicator for the irrigation quotas and N applications (Table 5). The maximum irrigation water and N application for yield, N *PFP*, and *EB* were 2100 m³·hm⁻² and 110 kg·hm⁻², respectively, and the maximum irrigation water and N application for *iWUE* were 1200 m³·hm⁻² and 270 kg·hm⁻², respectively. The maximum irrigation water and N application for yield, N *PFP*, and *EB* were maximized, whereas the maximum irrigation water and N application for irrigation water-use efficiency were maximized at the lowest level. Utilization efficiency was maximized with the lowest irrigation quota and largest N application.

**Table 5.** Maximum yield, water use efficiency, biased fertilizer productivity, and economic efficiency, and their corresponding irrigation and N application rates.

| Year | Irrigation Quotas m³/hm² | Nitrogen Fertilizer Amount Kg/hm² | Yield kg/hm² | *iWUE* kg/m³ | *PFP* kg/kg | EB CNY/hm² |
|---|---|---|---|---|---|---|
| 2021 | 2100.00 | 110 | 53,270.56 | 25.58 | 484.76 | 77,107.71 |
| | 1200.00 | 270 | 49,831.87 | 41.05 | 189.17 | 74,226.64 |
| | 2099.97 | 110 | 53,270.31 | 25.58 | 484.75 | 77,107.39 |
| | 2100.00 | 110 | 53,270.56 | 25.58 | 484.76 | 77,107.71 |
| 2022 | 2100.00 | 110 | 53,163.64 | 25.51 | 481.83 | 76,926.19 |
| | 1200.00 | 270 | 49,272.90 | 40.79 | 185.26 | 73,271.9 |
| | 2099.97 | 110 | 53,163.42 | 25.51 | 481.83 | 76,925.93 |
| | 2100.00 | 110 | 53,163.64 | 25.51 | 481.83 | 76,926.19 |

### 3.4. Optimization of Efficient Water and Nitrogen Regulation System for Potato in the Central Dry Zone of Ningxia Based on the TOPSIS Model

To further reflect the role of water and nitrogen regulation on potato yield, *iWUE*, nitrogen *PFP*, and economic efficiency, the entropy weight TOPSIS comprehensive evaluation model was used to seek the most efficient irrigation water and nitrogen fertilizer program for potato in the central Ningxia arid zone. Table 6 shows the score rankings for each treatment and also shows that the W3N1 treatment has the highest total score ranking, and the CK has the lowest total score ranking. This experiment recommended the W3N1 treatment as a better water and nitrogen coupling model for potato cultivation in the central dry zone of Ningxia.

**Table 6.** Statistics of TOPSIS judging indicators.

| Year | Treatments | $D_i^+$ | $D_i^-$ | $Ri$ | Rank | Year | Treatments | $D_i^+$ | $D_i^-$ | $Ri$ | Rank |
|------|-----------|--------|--------|------|------|------|-----------|--------|--------|------|------|
| 2021 | W1N1 | 0.1942 | 0.2737 | 0.585 | 3 | 2022 | W1N1 | 0.254 | 0.2162 | 0.4599 | 4 |
| | W1N2 | 0.3018 | 0.224 | 0.426 | 5 | | W1N2 | 0.3545 | 0.0882 | 0.1991 | 9 |
| | W1N3 | 0.3499 | 0.2679 | 0.4336 | 4 | | W1N3 | 0.411 | 0.0745 | 0.1535 | 10 |
| | W2N1 | 0.1808 | 0.2913 | 0.617 | 2 | | W2N1 | 0.1956 | 0.2734 | 0.5829 | 2 |
| | W2N2 | 0.3006 | 0.1785 | 0.3725 | 6 | | W2N2 | 0.311 | 0.1243 | 0.2856 | 7 |
| | W2N3 | 0.3612 | 0.1801 | 0.3327 | 7 | | W2N3 | 0.3763 | 0.0938 | 0.1996 | 8 |
| | W3N1 | 0.2115 | 0.3605 | 0.6302 | 1 | | W3N1 | 0.153 | 0.3551 | 0.6988 | 1 |
| | W3N2 | 0.3485 | 0.1082 | 0.237 | 10 | | W3N2 | 0.2658 | 0.1878 | 0.4141 | 5 |
| | W3N3 | 0.3837 | 0.1238 | 0.2439 | 9 | | W3N3 | 0.3386 | 0.1615 | 0.3229 | 6 |
| | CK | 0.3873 | 0.1506 | 0.2801 | 8 | | CK | 0.2423 | 0.2815 | 0.5374 | 3 |

Note: Si* and Si are the Euclidean distances, and Ci* is the relative closeness.

## 4. Discussion

Ningxia is located in the northwest arid inland region of China, and the natural water resources in this area have been limited in recent years. Drought and water shortages are the main causes of low and unstable potato yields in the region, causing tuber deformation. High nitrogen fertilizer applications increase input costs and also reduce the yield. Therefore, exploring a reasonable irrigation system and nitrogen application strategy could provide both theoretical and technical references to increase potato yield and improve quality.

Plant height is an important indicator of potato growth, and the crop stem is the main carrier of nutrients and water. It also supports the stability of leaves, flowers, and fruits, as well as photosynthesis and nutrient storage [32,33]. The results of this study showed that the irrigation quota and amount of nitrogen applied significantly affect plant height and stem thickness and that the stem thickness at the seedling stage initially increased and then decreased with the amount of nitrogen applied; plant height showed a gradual increase with the amount of irrigation and nitrogen applied. Notably, high irrigation and nitrogen application were not conducive to increased plant stem thickness, and this highlights the importance of the reasonable use of nitrogen fertilizer [34]. Overall, potato plant height and stem thickness were sensitive to the amount of water and nitrogen supplied at all fertility stages and varied considerably under field conditions. Moreover, irrigation had a greater impact on potato growth compared to N application.

Improved yield and quality are two of the main objectives of water and nitrogen regulation [35]; the average yield of potato-converted hectares was 39,931.68−54,085.48 kg/hm$^2$, with an average increase of 2139.35−16,293.14 kg/hm$^2$ compared to that under the CK, equivalent to an average increase of 43.11−66%. Potato yield was the highest under the W3N1 treatment; the N2 treatment showed a trend of increasing and then decreasing with irrigation water, and potato yield increased with the increase in irrigation [30]. Potato quality in terms of VC, reducing sugar, and starch contents improved under the different treatments compared with that under the CK, and the highest values for these parameters were noted under the W3N2, W1N2, W3N3, and W3N2 treatments, showing

an increase of 7.00, 50.00, and 33.00%, respectively. Moreover, increased irrigation and nitrogen application had a positive effect on yield [36].

Irrigation water-use efficiency is an important indicator for evaluating the efficiency of agricultural irrigation water use in irrigation areas [37–40]. *iWUE* was greatest under the W1N3 treatment, as the irrigation quota is an important parameter affecting irrigation water use efficiency [41]. *PFP* is an important indicator of the combined effects of local soil-base nutrient levels and fertilizer dosage [42]. Potato nitrogen *PFP* was higher in the same irrigation quota under the N1 treatment than that under the other treatments, and the highest and lowest nitrogen *PFP* were observed under the W3N1 and W1N3 treatments, respectively. This was because lower N applications and higher irrigation rates increased the N *PFP*, and the N *PFP* gradually increased with increasing N application at the same irrigation rate [43]. Economic efficiency is a key index for comprehensively evaluating and reflecting the technical and economic feasibility of planting. Unreasonable fertilizer applications reduce the economic efficiency and yield-to-input ratio of potatoes [44]. The highest economic efficiency was achieved under the W3N1 treatment, with a 42.46% increase compared with the CK; therefore, controlling the N fertilizer application amount is an important agronomic measure to improve potato yield, economic efficiency, and N fertilizer utilization. Our results suggest that water is the main factor for increasing yield in the arid areas of central Ningxia; however, different fertilizer application rates can be utilized to achieve better results. Obtaining high tuber yields in semi-arid areas without adequate water supply is extremely challenging. Other possibilities that can be investigated to optimize irrigation water and chemical fertilizer use include adopting new genotypes characterized by more efficient responses to irrigation and fertilization and which have been improved for earlier tuberization, higher harvest index, and better sink/source balance. The water and fertilizer saved may be used more profitably to irrigate and fertilize supplemental lands in a manner that supports more efficient and rational land use from an economic as well as environmental perspective [45].

Many scholars have investigated the reciprocal effects of yield, *iWUE*, N and *PFP*, and economic efficiency using regression methods and other models [46–48], which showed that the $R^2$ of the binary quadratic regression equation established by each indicator with irrigation quota and nitrogen application was > 0.85, which could better predict the effects of the changes in water and nitrogen regulation on each indicator. The optimal irrigation water and N application rates for better yield, N fertilizer bias productivity, and economic efficiency were 2100 $m^{-3}$ $hm^{-2}$ and 110 $kg^{-1}$ $hm^{-2}$, respectively, and the optimal irrigation water and N application rates for irrigation water use efficiency were 1200 $m^{-3}$ $hm^{-2}$ and 110 $kg^{-1}$ $hm^{-2}$, respectively. TOPSIS is a widely used evaluation method [49], and to further illustrate the optimal solution, the present study concluded that the W3N1 treatment (with irrigation at 1200 $m^3 \cdot hm^{-2}$ and N application at 110 $kg \cdot hm^{-2}$) was the most efficient treatment based on TOPSIS. This treatment can be used as an improved water and nitrogen control model for potato cultivation in the central arid zone of Ningxia.

## 5. Conclusions

Under the same conditions of irrigation, the variation patterns for stem thickness in the seedling stage showed a trend of increasing and decreasing with the increase in nitrogen application; however, plant height did not show a significant trend at the seedling stage under the W1 treatment. Under the same conditions of nitrogen application, the stem thickness initially decreased and subsequently increased with the increase in irrigation quota at each fertility stage. Plant height initially decreased and then increased with the increase in irrigation quota under each treatment at the seedling and tuber growth stages.

Under the same irrigation conditions, the W1 and W2 treatments showed an increasing trend with increasing N application, and the W3 treatment showed a decreasing trend followed by an increasing trend. The N1 and N3 treatments showed an increasing trend with increasing N application, and the N2 treatment showed an increasing trend followed

by a decreasing trend. The VC, reducing sugar, and starch contents of potatoes were highest under the W3N2, W1N2, and W3N3 treatments.

Irrigation water use efficiency tended to decrease with decreasing irrigation quotas at different N application rates. Potato N fertilizer bias productivity was higher under the N1 treatment than that under other treatments with the same irrigation quotas, and showed an increase under the N1 and N2 treatments and an increase and a subsequent decrease under the N3 treatment with the same irrigation quotas.

The binary quadratic regression equation for water and nitrogen was established with the objectives of yield, N fertilizer productivity, economic efficiency, and irrigation water use efficiency, and we concluded that water and nitrogen could significantly improve potato yield, N fertilizer productivity, and economic efficiency. The effect of irrigation water was greater than that of N application, and we observed a significant interaction between water and nitrogen. A regression equation was used to obtain the optimal water and nitrogen combination for different objectives in the dry zone of Ningxia.

The TOPSIS model was used for comprehensive analysis under water and nitrogen regulation test conditions. Irrigation quota and nitrogen application rates of 2100 $m^3 \cdot hm^{-2}$ and 110 $kg \cdot hm^{-2}$, respectively, could ensure optimal yield, quality, and economic benefits in potato growing areas in the central Ningxia arid zone, thus achieving the goals of high yield, high quality, and water and fertilizer conservation.

**Author Contributions:** Conceptualization: Y.Y.; methodology, formal analysis, funding acquisition, J.Y.; investigation, Z.M.; data curation, X.W.; data curation, F.S.; validation, Z.Y. All authors involved in this study helped in the writing and improvement of the manuscript. All authors have read and agreed to the published version of the manuscript.

**Funding:** This research was funded by the Ningxia First-class Discipline Construction Funding Project for Higher Education Institutions (Grant No. NXYLXK2021A03) and the Yunnan Applied Basic Research Projects (2019FD105).

**Institutional Review Board Statement:** Not applicable.

**Informed Consent Statement:** Not applicable.

**Data Availability Statement:** The data presented in this study are available within the article.

**Acknowledgments:** The authors would like to thank Wenqian Wang from for providing language editing services.

**Conflicts of Interest:** The authors declare no conflict of interest.

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
