# Peer review of "Water and Nitrogen Regulation Effects and System Optimization for Potato (Solanum tuberosum L.) under Film Drip Irrigation in the Dry Zone of Ningxia China"

_agronomy, doi:10.3390/agronomy13020308_

Round 1

Reviewer 1 Report

comments

Author Response

Response to Reviewer 1 Comments

Dear Editor, Dear reviewers

We would like to thank you very much for your valuable comments and good suggestions that greatly helped to improve our manuscript. Thank you very much for your time and efforts. Based on the comments and suggestions, we have revised the manuscript accordingly. And we have uploaded a version of the revised manuscript with all the changes highlighted by using the track changes mode in MS word and a version of the revised manuscript with all changes accepted.

Appended to this letter is our point by point response to the comments raised by the reviewers. The comments are reproduced and our responses are given directly afterward in a different color (red). The line number in Reply refers to the revised manuscript with track changes.

Point 1: There are many grammatical, punctuation, syntax errors, so sever English language editing is needed.

Response 1: Thank you for the kind advice. The paper has been grammatically checked for all content and edited in strict English in accordance with editorial requirements.

Point 2: Material and method Lacks other references

Response 2: Thank you for the kind advice. References have been added to support the "Materials and Methods" section and are detailed in references [27], [28], and [29, 30].

Point 3: For economical benefits , please provide the references and from line 171 please check ref 3, 5 and 6 and make explanation because the equation is unclear .

Response 3: Thank you for the kind advice. Economic efficiency has provided references [29, 30], where the economic efficiency is the cost of the total potato output gained has to be subtracted from the total inputs, including input irrigation water, applied nitrogen inputs, other inputs, and labor costs.

Equations 5 and 6 have been adjusted in the article.

Point 4: Why didn't you assess the quality of the potato in terms of size, depth, shape, and specific gravity?

Response 4: Thank you for the kind advice. The article assesses potato quality in terms of yield, irrigation water use efficiency, nitrogen and fertilizer bias, and economic income based on the principle of "yield-dependent water and yield-dependent fertilizer".

Point 5: Discussion requires scientific explanation

Response 5: Thank you for the kind advice. This paper studies the effects of reasonable water and nitrogen regulation on potato growth, yield, quality, water use efficiency, nitrogen and fertilizer bias productivity and economic benefits based on local conditions; constructs a mathematical model between water and nitrogen regulation and each index based on a binary quadratic regression model, and carries out program optimization, and analyzes each index of potato to determine the optimal amount of irrigation water and nitrogen application through TOPSIS model, so as to provide a scientific basis and practical guidance for high yield, water and fertilizer conservation in Ningxia dry zone. This study will provide scientific theoretical basis and practical guidance for high yield, water and fertilizer saving in potato cultivation under drip irrigation in dry areas of Ningxia.

Point 6: The paper lacks novelty , as there are many papers that have discussed this point in more detail

Response 6: Thank you for the kind advice. References [45] were added in the discussion section to further illustrate the optimization of water and nitrogen regimes for potato in the dry zone of Ningxia, China by water fertilization to end. With see the red marked part of the article.

We appreciate for Editor and Reviewers' warm work earnestly, and hope that the correction will meet with approval. Once again, thank you and all the reviewers for the advice and consideration.

Sincerely,

Dr.YingPan Yang

College of Civil and Hydraulic Engineering,

Ningxia University,

Yinchuan, 750021, China

E-mail: 12021140105@stu.nxu.edu.cn

Reviewer 2 Report

Manuscript title:  Water and Nitrogen Regulation Effects and System Optimization for Potato (Solanum tuberosum L.) under Film Drip Irrigation in the Dry Zone of Ningxia China

Authors:  Yingpan Yang et al.

The research issues of the manuscript are up-to-date and concern the optimization of irrigation and nitrogen fertilization of potatoes grown in a region prone to drought, which justifies the research factors used. The research results obtained are well illustrated with clear figures and tables, and the statistical interpretation of the results is at an appropriate level.

However, the following points should be considered to improve this work:

Introduction: please cite 1-2 publications concerning irrigation and nitrogen fertilization of potatoes in general, and not only other species (rice, wheat, cotton, etc.),

Materials and Methods

Line 97: meteorological conditions - please specify the years of long-term weather data,

Line 101-105: methods of determination must be provided, please provide references of the methods,

Line 119-120: nitrogen fertilizer? please tell me the name of the nitrogen fertilizer, how much % nitrogen?

Line 125: the potato variety “green potato nine”, is this variety popular and is it registered for cultivation in China?

Line 127: seeds were sown..? Is this the correct notation? Potatoes in the experiment were planted (tubers) or sown (seeds),

Line 134-135: please briefly give the most important agrotechnical data of potato cultivation, forecrop, tillage, pesticides, doses, etc.

Conclusions: Conclusions very extensively written.., I believe there are other a lot of nice conclusions could be made from this manuscript

Author Response

Response to Reviewer 2 Comments

Dear Editor, Dear reviewers
We would like to thank you very much for your valuable comments and good
suggestions that greatly helped to improve our manuscript. Thank you very much for your time and efforts. Based on the comments and suggestions, we have revised the manuscript accordingly. And we have uploaded a version of the revised manuscript with all the changes highlighted by using the track changes mode in MS word and a version of the revised manuscript with all changes accepted.

Appended to this letter is our point by point response to the comments raised by the reviewers. The comments are reproduced and our responses are given directly afterward in a different color (red). The line number in Reply refers to the revised manuscript with track changes.

Point 1: Introduction: please cite 1-2 publications concerning irrigation and nitrogen fertilization of potatoes in general, and not only other species (rice, wheat, cotton, etc.),

Response 1: Thank you for the kind advice. Three papers on potato irrigation and nitrogen fertilization have been cited. For details see articles [16] and [17].

Point 2: Line 97: meteorological conditions - please specify the years of long-term weather data,

Response 2: Thank you for the kind advice. The year in which the modification of meteorological data has been carried out. Details" with an average rainfall of about 270 mm and annual evaporation of about 2,325 mm from 2012 to 2022,".

Point 3: Line 101-105: methods of determination must be provided, please provide references of the methods,

Response 3: Thank you for the kind advice. References for the assay methods have been provided. Details are given in the article reference [24].

Point 4: Line 119-120: nitrogen fertilizer? please tell me the name of the nitrogen fertilizer, how much % nitrogen?

Response 4: Thank you for the kind advice. The nitrogen fertilizer used in this experiment was 46% urea. It has been modified in the text.

Point 5: Line 125: the potato variety “green potato nine”, is this variety popular and is it registered for cultivation in China?

Response 5: Thank you for the kind advice. The potato variety “qingshu9” was tested in the experiment, "qingshu9" is a potato variety selected by the Institute of Biotechnology, Qinghai Academy of Agricultural and Forestry Sciences with 3875213xAPHRODITE, which is drought-resistant, cold-resistant, resistant to late blight and ring rot.

Point 6: Line 127: seeds were sown..? Is this the correct notation? Potatoes in the experiment were planted (tubers) or sown (seeds),

Response 6: Thank you for the kind advice. Potatoes are grown as tubers and are "seed potatoes", which has been changed in the text.

Point 7: Line 134-135: please briefly give the most important agrotechnical data of potato cultivation, forecrop, tillage, pesticides, doses, etc.

Response 7: Thank you for the kind advice. Revised in the thesis. See the paper for details. “Select detoxified seed potatoes before planting, firstly, when selecting seed potatoes, eliminate old potatoes, deformed potatoes, damaged potatoes, diseased potatoes and rotten potatoes, germinate qualified seed potatoes, put seed potatoes in warm indoor conditions before sowing, the germination length is 0.5-1 cm; secondly, cut seed pota-toes, seed potatoes should be cut 2-3 d before sowing, ensure that there are 1-2 bud eyes on each cut block, generally the cut block should not be The cutter used for cutting should be disinfected with 75 % alcohol to prevent the spread of diseases during the cutting process. Finally, the seeds are mixed with pharmaceuticals to prevent late blight, bacteria and semi-intelligent fungi, so when choosing pharmaceuticals, we should choose the pharmaceuticals to prevent late blight, bacteria and semi-intelligent fungi respectively.”

Point 8: Conclusions: Conclusions very extensively written.., I believe there are other a lot of nice conclusions could be made from this manuscript.

Response 8: Thank you for the kind advice. Revised in the thesis. See the paper for
details.

We appreciate for Editor and Reviewers' warm work earnestly, and hope that the
correction will meet with approval.
Once again, thank you and all the reviewers for the advice and consideration.
Sincerely,
Dr.YingPan Yang

College of Civil and Hydraulic Engineering,
Ningxia University,
Yinchuan, 750021, China
E-mail: 12021140105@stu.nxu.edu.cn

Round 2

Reviewer 1 Report

thank you for responding and i have no more comments